

# Health-related quality of life and its associated factors among Chinese seasonal retired migrants in Hainan

Sikun Chen[1], Tianchang Li[2], Lingjun Wang[2], Shigong Wang[3], Lin Ouyang[4], Jiwei Wang[1], Dayi Hu[2,5] and Jinming Yu[1]

[1] School of Public Health, Fudan University, Shanghai, China
[2] The Second People's Hospital of Hainan Province, Wuzhishan, Hainan, China
[3] School of Atmospheric Sciences, Chengdu University of Information Technology, Chengdu, Sichuan, China
[4] WUZHISHAN Snowbird Medical Professionals Workstation, Wuzhishan, Hainan, China
[5] People's Hospital of Peking University, Beijing, China

Corresponding authors
Dayi Hu, hudayihust@126.com
Jinming Yu, jmy@fudan.edu.cn

## ABSTRACT

**Background.** Seasonal retired migrants are mainly retired or semiretired people who migrate to warmer areas during the winter and then return to their original homes in the following year. Despite its increasing popularity, the evidence concerning the health status of Chinese seasonal retired migrants is controversial. Although some studies have shown that seasonal retired migrants have better health status, other studies note that migrants are driven involuntarily by health concerns and that their mental health suffers during migration. The objective of this study was to provide quantified evidence on the health-related quality of life (HRQoL) of seasonal retired migrants in China and to identify potential factors associated with HRQoL.

**Methods.** This was a community-based cross-sectional study of seasonal retired migrants who lived in Wuzhishan, Hainan during the winter. The Chinese version of the EuroQol 5-Dimension 5-Level (EQ-5D-5L) was used to assess participants' HRQoL. Data on demographic and behavioral characteristics, body mass index, and the presence of chronic diseases were collected using a self-administered survey. Logistic regressions were used to explore the factors associated with responses in each dimension of the EQ-5D-5L descriptive system, and a multiple linear regression model was used to explore the factors associated with EuroQol visual analogue scale (EQ-VAS) scores.

**Results.** A total of 992 participants (female: 56.5%) were enrolled in the current study, with a mean age of $67.87 \pm 6.98$ years. Most participants reported problems in the pain/discomfort dimension (16.7%), followed by the anxiety/depression dimension (8.3%). Few participants reported problems in the first three dimensions of the EQ-5D-5L: mobility (5.4%), usual activities (2.0%), and self-care (1.2%). The median EQ-VAS score was 84 (interquartile range: 80–89). The regression results indicated that poor sleep quality, the presence of chronic diseases, and low-level physical activity were important factors that were negatively associated with multiple dimensions of the EQ-5D-5L. In addition, the EQ-VAS score was negatively associated with age, the presence of chronic diseases, poor sleep quality, and low-level physical activity.

**Conclusion.** This study revealed that Chinese seasonal retired migrants have high HRQoL. In addition, this study revealed that sleep quality and physical activity are

correlated with multiple dimensions of the EQ-5D-5L descriptive system and EQ-VAS. Therefore, lifestyle interventions related to sleep quality and physical activity are essential for improving HRQoL among Chinese seasonal retired migrants.

## INTRODUCTION

Every winter, many retired or semiretired people move to southern provinces to escape cold weather in their original locations and then return to their places of origin in the following year. These seasonal retired migrants are attracted mostly to warm weather and are occasionally referred to as 'snowbirds' given their similarity to migratory birds (*Krout, 1983*). Seasonal retired migration is enabled by a privileged socioeconomic status (*O'Reilly, 2000*), and most seasonal retired migrants in China are generally from middle-class backgrounds (*Wang et al., 2020*). Hainan Province is one of the most popular destinations for seasonal retired migrants in China (*Chen & Wang, 2020*). The seasonal migrant population has increased continuously, and there are more than one million seasonal migrants each year during the winter in Hainan (*Hainan Provincial Bureau of Statistics, 2020*).

The health status of seasonal retired migrants has gained the attention of many researchers. A study based on health administrative databases revealed that seasonal retired migrants are healthier and more socioeconomically advantaged than non-migrants in their original locations (*Shariff et al., 2021*). Studies on Canadian and American older seasonal migrants have also revealed their high levels of self-rated health (*Fortuna, 2024*; *Kelly, 2023*). In China, some qualitative studies (*Wang, Cui & Xu, 2024*; *Zhou et al., 2018*) have revealed seasonal retired migrants' views on the potential positive effects of seasonal migration on health and their high levels of self-perceived health status. The superior health status of seasonal retired migrants could be attributed to the well-known theories of the "healthy migrant effect" (*Huang et al., 2023*; *Zhong et al., 2013*) and "salmon bias" (*Chen et al., 2022*). The former refers to the phenomenon in which healthier individuals are more likely to migrate as they can overcome the rigor of migration, and the latter theory refers to unhealthy migrants having a greater tendency to migrate back to their original communities than healthy migrants do.

However, contrary to the "healthy migrant effect" theory, *Kou, Xu & Hannam (2017)* reported that the health of some Chinese seasonal retired migrants deteriorates before they leave their original locations. They reported that Chinese seasonal retired migrants are mainly involuntarily driven by health conditions, although they are more willing to stay in their original locations. These seasonal retired migrants found it unbearable to experience the winter in their locations of origin. Moreover, seasonal retired migrants may encounter difficulties adjusting their psychological feelings at destinations while maintaining strong social and emotional connections with their places of origin, which could be the reason
pulling them back to their original places in the following year (*Kou, Xu & Kwan, 2018*). In addition, China's place-bound welfare systems can create barriers, as seasonal retired migrants may not be able to readily use their medical insurance at their destinations. Furthermore, they often have limited confidence in the quality of health care services in Hainan Province (*Chen & Wang, 2020*). Thus, although seasonal migration may improve physical health, the migrants' psychological health might not experience the same benefits and may even worsen.

Given that most current studies on Chinese seasonal retired migrants' health status are qualitative and their findings are controversial, further quantitative studies measuring Chinese seasonal retired migrants' health status are needed to gain a more comprehensive understanding. Health-related quality of life (HRQoL), which can quantitatively measure individuals' self-perceived health status in multiple dimensions (*Karimi & Brazier, 2016*), has been widely used as a health outcome in clinical research (*Andresen & Meyers, 2000*). In addition, HRQoL is widely used to assess older adults' health status in community settings (*Halaweh et al., 2015*; *Olivares et al., 2011*). Although few studies have incorporated HRQoL to determine the health status of seasonal retired migrants, HRQoL has been used in studies of migrant workers (*Cho, Kang & Park, 2023*). With circular migration patterns akin to those of seasonal retired migrants, the HRQoL of Chinese rural-urban migrant workers is comparable, and not inferior to, that of non-migrant rural residents (*Dai et al., 2015*; *Li et al., 2020*).

The determinants of HRQoL among Chinese internal migrants have been well documented. Demographic variables, including sex, marital status, financial status, and education, together with medical conditions, are strongly associated with HRQoL (*Dai et al., 2015*). In addition, unhealthy behaviors were also found to be associated with adverse health conditions (*Yang et al., 2017*). However, HRQoL and its associated factors have yet to be researched adequately among Chinese seasonal retired migrants.

Thus, this study sought to assess the HRQoL of Chinese seasonal retired migrants in Hainan and to explore the factors associated with migrants' HRQoL. In this study, on the basis of the results of previous research on Chinese internal migrants, we hypothesized that several important characteristics, including sex, age, marital status, education level, income level, presence of chronic diseases, body mass index (BMI), and several behaviors, are associated with the HRQoL of Chinese seasonal retired migrants. The findings of the present study could be useful for policy-makers to design tailored policies for improving Chinese seasonal retired migrants' health.

# MATERIALS & METHODS

## Study setting and design

With local residents of approximately 111.2 thousand people, Wuzhishan city is located in the central area of Hainan Island at an altitude of approximately 300 m, covering 1,144 km$^2$. Owing to the low latitude, the ambient temperature during the winter in Wuzhishan city is much higher than that in high-latitude regions. In addition, approximately 75% of the areas in Wuzhishan city are covered by forests, providing fresh and clean air. Owing to the

superior environment, a substantial population of seasonal retired migrants, most of whom reside in urban areas, move to Wuzhishan city in winter. According to a report from the *Hainan Health Commission (2019)*, the population of seasonal retired migrants in winter can reach approximately one-third of the local resident population in Wuzhishan city. Hence, this city is an ideal location for studying seasonal retired migrants. We conducted this community-based cross-sectional survey in the urban areas of Wuzhishan city from October to November 2022.

## Study setting and participants

The current study focused on seasonal retired migrants aged 50 years and older who were recruited from communities in the urban areas of Wuzhishan city, Hainan. The seasonal retired migrants were snowbirds who spend winter in Wuzhishan city, with a length of stay ranging from one month to eleven months per year. Sunbirds that escaped summer in their original places and did not overwinter in Wuzhishan city were excluded from this study because they represented a small proportion of the seasonal migrants in Wuzhishan city and were different from snowbirds in terms of their characteristics.

The inclusion criteria of this study were as follows: (1) aged 50 years and older; (2) owned or rented residences in Wuzhishan city; (3) had a stay of at least 30 consecutive days, which is consistent with previous studies on seasonal retired migrants (*Happel & Hogan, 2002*; *Smith & House, 2006*); and (4) had experienced at least two rounds of seasonal migration.

The exclusion criteria were as follows: (1) living in Wuzhishan city for more than eleven months per year to distinguish our participants from local permanent residents; (2) arriving in Wuzhishan city between March and August and left between September and December to exclude sunbirds who did not spend winter in Wuzhishan city; (3) refusal to complete the questionnaire; and (4) unable to complete the questionnaire independently or with help from the interviewers.

We used the following formula (*Desu & Raghavarao, 1990*) to calculate the sample size: $n = (Z_{\alpha/2} * \sigma/d)^2$, where $n =$ the desired sample size, $Z_{\alpha/2} =$ the upper $100(\alpha/2)$ percentile point of the standard normal distribution, $\sigma =$ the assumed population standard deviation, and $d =$ the desired precision. In this study, the confidence level was set to 0.95; hence, $Z_{\alpha/2} = 1.96$. The assumed population standard deviation was 1.53 according to the standard deviation of EuroQol-visual analogue scale (EQ-VAS) scores in the norms in the urban Chinese population (*Yang et al., 2018*), resulting in $\sigma = 1.53$. The desired precision was set to one tenth of $\sigma$; hence, $d = 0.15$. The minimum sample size for the current study was 400.

To select a representative sample, we utilized a cluster sampling method to randomly select participants for our study. We randomly selected 10 communities in the urban areas of Wuzhishan city. Within each community, the local neighborhood committee, an organization that is responsible for residential management at the grassroots level, identified potential seasonal retired migrants based on their seasonal movements. The potential seasonal retired migrants were further screened on the basis of the predefined inclusion and exclusion criteria by trained interviewers.

Eligible participants were then invited to the local study site, where trained interviewers first instructed them about the study's aim. The participants were then required to complete self-administered questionnaires. Those who encountered problems reading or understanding the questionnaires could receive help from interviewers. All completed questionnaires were assessed onsite by interviewers to ensure that there were no missing values. If any missing items were identified, the participant was required to complete the missing items instantly. As a result, there were no missing values in the current study.

## Variable measurements
### Outcome variables

The outcome variable for this study is HRQoL, which reflects participants' self-perceived physical and mental health. We adopted the EuroQol 5-Dimension 5-Level (EQ-5D-5L) scale to assess participants' HRQoL (*Feng et al., 2021*). The EQ-5D-5L instrument was originally developed by the EuroQol Group and is available in Chinese (*Luo et al., 2013*), and is widely used in epidemiological studies targeting the general population (*Tan et al., 2013*; *Wong et al., 2021*).

The EQ-5D-5L instrument consists of an EQ-5D-5L descriptive system and the EQ-VAS. The EQ-5D-5L descriptive system measures participants' health in five dimensions: mobility, self-care, usual activities, pain/discomfort, and anxiety/depression. Each dimension has five levels: no problems, slight problems, moderate problems, serious problems, and extreme problems. The EQ-VAS measures participants' self-rated health, ranging from 0 to 100, where 0 indicates the participants' worst imaginable health status and 100 indicates the best health status imaginable.

For the psychometric property of the EQ-5D-5L, according to a systematic review (*Feng et al., 2021*), nine studies reported intraclass correlation coefficients (ICCs) of at least 0.70, indicating good to excellent test-retest reliability, which means that the EQ-5D-5L could produce similar results in stable environments. This review (*Feng et al., 2021*) also revealed a pooled Spearman coefficient of 0.756 between the EQ-5D-5L and multiattribute utility instruments, indicating excellent convergent validity. This finding means that the EQ-5D-5L could accurately capture participants' HRQoL. Hence, the EQ-5D-5L is a reliable and valid tool for estimating Chinese seasonal retired migrants' HRQoL.

### Explanatory variables

This study considered the following factors as explanatory variables: demographic characteristics, behavioral factors, BMI, and the presence or absence of chronic diseases.

The participants' demographic characteristics investigated were sex, age, current marital status, education level and income level. The participants' current marital status was categorized as married or unmarried. Education level was categorized as primary school or lower, middle school, high school, or university or above. Income level was determined according to participants' household monthly income per capita (*i.e.*, the total gross household income monthly income divided by the total number of family members) and categorized into three levels (<5000 CNY, 5000-9999 CNY, or $\geq$10000 CNY).

The behavioral factors selected in the current study included smoking status, alcohol consumption, physical activity, and sleep quality. Smoking status and alcohol consumption were categorized into three levels (never, ceased, current).

Physical activity was measured using the International Physical Activity Questionnaire (IPAQ) (*Craig et al., 2003*) developed by an International Consensus Group with the support of the World Health Organization and the U.S. Centers for Disease. Available in Chinese, the IPAQ has two versions: the long (31 items) and short (nine items) forms. We adopted the short form of IPAQ, which asks the participants to report the frequency and duration of walking, and all vigorous and moderate activities lasting at least 10 min. According to IPAQ guidelines (*I.P.A.Q. Research Committee, 2005*), the data could be converted to metabolic equivalent scores (MET min wk$^{-1}$) and classified into three categories, including low, moderate, and high. A participant with a high physical level should meet any one of the following two criteria: (1) vigorous intensity activity on at least three days and an accumulation of at least 1500 MET min wk$^{-1}$, or (2) seven or more days of any combination of walking, moderate intensity or vigorous intensity activities achieving a minimum of at least 3000 MET min wk$^{-1}$. A participant with moderate physical level should meet any one of the following three criteria and not meet the criteria for high physical level: (1) three or more days of vigorous activity of at least 20 min per day, (2) five or more days of moderate intensity activity or walking of at least 30 min per day, or (3) five or more days of any combination of walking, moderate intensity or vigorous intensity activities achieving a minimum of at least 600 MET min wk$^{-1}$. Individuals who did not meet the criteria for moderate or high physical activity were considered to have a low level of physical activity (*I.P.A.Q. Research Committee, 2005*). The Chinese version of the IPAQ-Short Form was found to have ICCs ranging from 0.81 to 0.89, indicating good retest reliability. The physical activity measured by the IPAQ was moderately related to the pedometer data (Spearman $r = 0.33$), supporting the acceptable validity of the Chinese version of the IPAQ-Short Form.

Sleep quality was measured with the Pittsburgh Sleep Quality Index (PSQI), developed by *Buysse et al. (1989)*. The PSQI is a self-rating questionnaire with 19 questions. The global PSQI score ranges from 0 to 21, with higher scores indicating poorer sleep quality. The threshold of the Chinese PSQI is 7 (*i.e.,* an individual with a PSQI score >7 has poor sleep quality) (*Liu et al., 1996*). The Chinese PSQI has a retest coefficient of 0.994, indicating excellent retest reliability. In addition, the PSQI was found to be highly related to other sleep quality assessment tools (correlation coefficient = 0.842), indicating high convergent validity (*Lu et al., 2014*).

In addition, BMI was calculated on the basis of participants' reported weights and heights and treated as a continuous variable in the analysis. The participants also reported whether they had chronic diseases.

### Ethical considerations

All participants included in this study provided written informed consent. Prior to participation, individuals were provided with detailed information about the study's objectives, procedures, potential risks and benefits and their rights to withdraw from

the study at any point without consequences. This study was approved by the medical ethics committee of the School of Public Health, Fudan University (IRB00002408 & FWA00002399, approval number IRB#2022-02-0944). All procedures in this study were performed in accordance with the ethical standards outlined in the Declaration of Helsinki.

## Statistical analysis

We provided frequencies and percentages for participants' demographic characteristics, presence of chronic diseases, categorical BMI, and behavioral factors to describe the participants' characteristics. In addition, we also provided the means and standard deviations for age and BMI. To describe participants' HRQoL, we provided frequencies and percentages for participants' responses to each dimension of the EQ-5D-5L descriptive system. The EQ-VAS scores were summarized using means and standard deviations and medians and interquartile ranges (IQRs).

When analyzing the factors associated with HRQoL, as described in previous studies (*Kim et al., 2018*; *Wong et al., 2021*), we first dichotomized the responses in each dimension of the EQ-5D-5L descriptive system into two levels: (1) reporting no problems and (2) reporting any problem (slight, moderate, severe or extreme problems). We then conducted separate logistic regression modeling of the probability of reporting any problem in each dimension with the same regressors, including selected demographic and behavioral characteristics, BMI, and the presence of chronic diseases. Because participants who were unmarried reported no problems in the dimension of self-care, leading to quasicomplete separation problems, the marital status variable was not included in the self-care model. Similarly, because participants with an education level of middle school or lower and those who were current smokers reported no problems in the self-care dimension, these participants were excluded from the self-care model. Odds ratios (ORs) were computed from the five logistic regression models. The five logistic regression models can be expressed as follows:

$$logit\left(P\left(Y=1\right)\right)$$
$$=\beta_0+\beta_1 Sex+\beta_2 Age+\beta_3 Marital+\beta_4 Education+\beta_5 Income+\beta_6 ChronicDiseases$$
$$+\beta_7 BMI+\beta_8 Smoking+\beta_9 Alcohol+\beta_{10} Sleep+\beta_{11} PhysicalActivity$$

Here,

- $P(Y=1)$ is the probability of the participant reporting any problem in the corresponding dimension of the EQ-5D-5L descriptive system.
- $\beta_0$ is the intercept, and $\beta_1$ to $\beta_{11}$ are the coefficients representing the effect of each independent variable on the log-odds of the outcome.

We also conducted a multiple linear regression model using the EQ-VAS score as the dependent variable and the same regressors used in the logistic regression models. The multiple linear regression model can be expressed as follows:

$$EQ-VASscore$$
$$=\beta_0+\beta_1 Sex+\beta_2 Age+\beta_3 Marital+\beta_4 Education+\beta_5 Income+\beta_6 ChronicDiseases$$

$$+\beta_7 BMI + \beta_8 Smoking + \beta_9 Alcohol + \beta_{10} Sleep + \beta_{11} PhysicalActivity$$

Here,

- $\beta_0$ is the intercept, and $\beta_1$ to $\beta_{11}$ are the coefficients representing the effect of each independent variable on the EQ-VAS score.

All the statistical analyses were performed by using SAS software version 9.4 (SAS Institute, Cary, NC, USA). A two-sided $p$ value of less than 0.05 was considered statistically significant.

## RESULTS

### Characteristics of participants

A total of 1069 eligible participants were invited to participate in the current study, 992 of whom completed the survey, yielding a response rate of 92.8%. The mean age of the participants was $67.87 \pm 6.98$ years, and approximately half of them (56.5%) were female. The majority of the participants (93.6%) were married. University or above was the most common level of education (36.7%), followed by high school (30.3%). Slightly more than half of the seasonal retired migrants (54.7%) had an income level of less than 5000 CNY per month. A total of 41.5% had chronic diseases. The mean BMI of the participants was $24.11 \pm 3.03$. The majority of the seasonal retired migrants had never smoked (89.4%) or drunk alcohol (62.5%). Details regarding the characteristics of the study population are presented in Table 1.

### Health-related quality of life

Table 2 shows the distribution of the participants' responses to each dimension of the EQ-5D-5L descriptive system and their EQ-VAS scores. Almost all of them reported no problems in the self-care dimension (98.8%) or usual activities dimension (98.0%), followed by the mobility dimension (94.6%) and anxiety/depression dimension (91.7%). The participants most commonly reported problems (slight to extreme problems) in the dimension of pain/discomfort (16.7%). In addition, the median EQ-VAS score was 84 (IQR: 80–89).

Table 3 presents the results of logistic models for each dimension of the EQ-5D-5L descriptive system. Sleep quality was the most important factor and was negatively associated with all five dimensions of the EQ-5D-5L descriptive system. In addition, the presence of chronic diseases was negatively associated with three dimensions of the EQ-5D-5L descriptive system (mobility, pain/discomfort, and anxiety/depression). Compared with those with low-level physical activity, participants with moderate-level physical activity were less likely to report problems in the pain/comfort and anxiety/depression dimensions. In addition to the pain/comfort and anxiety/depression dimensions, high-level physical activity was additionally associated with reporting fewer problems in the usual activities dimension. The responses to the EQ-5D-5L descriptive system were also associated with other factors, including education level, income level, smoking status, and alcohol consumption.

Table 4 presents the results of the multiple linear regression for the EQ-VAS score. Lower EQ-VAS scores were significantly associated with age ($\beta = -0.15$, $p = 0.001$), the

**Table 1** Characteristics of participants (*n* = 992).

| | n (%) |
|---|---|
| Total | 992 (100) |
| Sex | |
|     Male | 432 (43.6) |
|     Female | 560 (56.5) |
| Age | |
|     50–59 years | 121 (12.2) |
|     60–69 years | 490 (49.4) |
|     70–79 years | 325 (32.8) |
|     $\geq$80 years | 56 (5.7) |
| Current marital status | |
|     Married | 928 (93.6) |
|     Unmarried | 64 (6.5) |
| Education level | |
|     Elementary school or less | 52 (5.2) |
|     Middle school | 275 (27.7) |
|     High school | 301 (30.3) |
|     University or above | 364 (36.7) |
| Income level | |
|     <5000 CNY | 543 (54.7) |
| 5000–9999 CNY | 382 (38.5) |
|     $\geq$10000 CNY | 67 (6.8) |
| Presence of chronic diseases | |
|     Yes | 412 (41.5) |
|     No | 580 (58.5) |
| BMI | |
|     <18.5 | 22 (2.2) |
|     18.5–23.9 | 479 (48.3) |
|     24.0–27.9 | 393 (40.0) |
|     $\geq$28.0 | 98 (9.9) |
| Smoking status | |
|     Current | 58 (5.9) |
|     Ceased | 47 (4.7) |
|     Never | 887 (89.4) |
| Alcohol consumption | |
|     Current | 255 (25.7) |
|     Ceased | 117 (11.8) |
|     Never | 620 (62.5) |
| Sleep quality | |
|     Poor | 136 (13.7) |
|     Good | 856 (86.3) |
| Physical Activity | |
|     Low | 150 (15.1) |

**Table 1** (*continued*)

|  | n (%) |
|---|---|
| Moderate | 403 (40.6) |
| High | 439 (44.3) |

**Notes.**
BMI, Body mass index.

**Table 2  Distribution of EQ-5D-5L descriptive system and EQ-VAS scores among participants.**

| EQ-5D-5L | n (%) |
|---|---|
| Mobility |  |
| No problems | 938 (94.6) |
| Slight problems | 33 (3.3) |
| Moderate problems | 10 (1.0) |
| Severe problems | 8 (0.8) |
| Extreme problems | 3 (0.3) |
| Self-care |  |
| No problems | 980 (98.8) |
| Slight problems | 8 (0.8) |
| Moderate problems | 3 (0.3) |
| Severe problems | 1 (0.1) |
| Extreme problems | 0 |
| Usual activities |  |
| No problems | 972 (98.0) |
| Slight problems | 14 (1.4) |
| Moderate problems | 2 (0.2) |
| Severe problems | 4 (0.4) |
| Extreme problems | 0 |
| Pain/discomfort |  |
| No problems | 826 (83.3) |
| Slight problems | 134 (13.5) |
| Moderate problems | 23 (2.3) |
| Severe problems | 6 (0.6) |
| Extreme problems | 3 (0.3) |
| Anxiety/depression |  |
| No problems | 910 (91.7) |
| Slight problems | 74 (7.5) |
| Moderate problems | 7 (0.7) |
| Severe problems | 1 (0.1) |
| Extreme problems | 0 |
| EQ-VAS, mean (SD) | 82.7 (10.3) |
| EQ-VAS, median (IQR) | 84 (80–89) |

**Notes.**
EQ-5D-5L, EuroQol 5-Dimension-5-Level; EQ-VAS, EuroQol-visual analogue scale.

**Table 3** Multivariable logistic regression showing odds ratios estimates of reporting problems in each dimension of EQ-5D-5L descriptive system by participants' characteristics.

| | EQ-5D dimensions, OR (95%CI) | | | | |
|---|---|---|---|---|---|
| | Mobility | Self-care | Usual activities | Pain/discomfort | Anxiety/depression |
| Sex (ref: male) | 1.07 (0.54,2.13) | 1.92 (0.37,10.08) | 1.05 (0.35,3.16) | 1.39 (0.90,2.15) | 0.87 (0.50,1.54) |
| Age (years) | 1.03 (0.99,1.08) | 0.98 (0.89,1.09) | 1.01 (0.94,1.08) | 1.01 (0.98,1.04) | 1.01 (0.97,1.04) |
| Marital status (ref: married) | 0.74 (0.21,2.63) | Predicts perfectly | 0.92 (0.36,9.26) | 1.78 (0.91,3.50) | 0.70 (0.23,2.13) |
| Education (ref: high school) | | | | | |
|     Elementary school or less | 0.42 (0.05,3.44) | Predicts perfectly | 2.06 (0.35,11.99) | 0.69 (0.21,2.20) | 0.34 (0.04,2.75) |
|     Middle school | 0.94 (0.43,2.07) | Predicts perfectly | 0.27 (0.05,1.38) | 1.12 (0.65,1.92) | 0.94 (0.45,1.98) |
|     University or above | 1.03 (0.48,2.19) | 0.55 (0.13,2.29) | 0.69 (0.22,2.21) | 1.85 (1.12,3.03)* | 1.90 (0.99,3.64) |
| Income (ref: 5000–9999 CNY) | | | | | |
|     <5000 CNY | 1.35 (0.68,2.69) | 0.41 (0.07,2.49) | 0.87 (0.28,2.77) | 1.04 (0.67,1.61) | 0.95 (0.54,1.68) |
|     ≥10000 CNY | 2.00 (0.73,5.46) | 5.02 (1.25,20.22)* | 3.13 (0.83,11.73) | 1.77 (0.91,3.43) | 1.05 (0.43,2.52) |
| Presence of chronic diseases (ref: no) | 2.90 (1.51,5.59)** | 3.74 (0.71,19.57) | 2.09 (0.72,6.04) | 4.72 (3.12,7.14)*** | 3.29 (1.88,5.76)*** |
| BMI (kg/m²) | 1.03 (0.93,1.14) | 0.97 (0.78,1.22) | 1.03 (0.88,1.21) | 1.02 (0.96,1.09) | 0.96 (0.89,1.04) |
| Smoking status (ref: never) | | | | | |
|     Current | 2.21 (0.76,6.36) | Predicts perfectly | 0.59 (0.06,5.71) | 1.47 (0.66,3.31) | 1.53 (0.54,4.33) |
|     Ceased | 2.75 (1.03,7.33)* | 4.25 (0.48,37.81) | 1.38 (0.24,7.88) | 1.85 (0.87,3.92) | 1.88 (0.76,4.66) |
| Alcohol consumption (ref: never) | | | | | |
|     Current | 1.75 (0.91,3.39) | 1.47 (0.323,6.46) | 2.44 (0.83,7.16) | 1.88 (1.21,2.92)** | 0.96 (0.52,1.79) |
|     Ceased | 0.60 (0.19,1.86) | 0.64 (0.07,6.38) | 1.78 (0.43,7.31) | 1.37 (0.78,2.42) | 2.43 (1.27,4.65)** |
| Sleep quality (ref: good) | 2.49 (1.31,4.74)** | 4.00 (1.02,15.69)* | 3.44 (1.22,9.68)* | 2.02 (1.29,3.19)** | 2.32 (1.33,4.07)** |
| Physical Activity (ref: low) | | | | | |
|     Moderate | 0.85 (0.38,1.92) | 0.47 (0.09,2.43) | 0.48 (0.17,1.38) | 0.40 (0.25,0.66)*** | 0.41 (0.22,0.75)** |
|     High | 0.91 (0.40,2.05) | 0.59 (0.11,3.06) | 0.23 (0.06,0.82)* | 0.42 (0.25,0.68)*** | 0.34 (0.18,0.64)*** |

**Notes.**

EQ-5D-5L, EuroQol 5-Dimension-5-Level; OR, Odds ratio; BMI, Body mass index.

*$p < 0.05$

**$p < 0.01$

***$p < 0.001$

Participants who were unmarried, with an education level of middle school or lower, or current smokers reported no problems in the dimension of self-care. The variable current marital status was not included in the self-care model. Participants with an education level of middle school or lower, or who were current smokers were excluded from the self-care model.

presence of chronic diseases ($\beta = -2.24$, $p < 0.001$), and poor sleep quality ($\beta = -6.90$, $p < 0.001$). Compared with low-level physical activity, moderate-level physical activity ($\beta = 5.01$, $p < 0.001$) and high-level physical activity ($\beta = 5.35$, $p < 0.001$) were associated with higher EQ-VAS scores.

**Table 4  Results of multiple linear regression for EQ-VAS scores.**

| Predictor | β | 95%CI | p |
|---|---|---|---|
| Sex (ref: male) | −0.19 | (−1.58, 1.19) | 0.787 |
| Age (years) | −0.15 | (−0.24, −0.06) | 0.001[**] |
| Marital status (ref: married) | 1.53 | (−0.97, 4.03) | 0.230 |
| Education (ref: high school) | | | |
|     Elementary school or less | 1.10 | (−1.85, 4.05) | 0.464 |
|     Middle school | −0.49 | (−2.11, 1.13) | 0.553 |
|     University or above | −0.32 | (−1.95, 1.30) | 0.695 |
| Income (ref: 5000–9999 CNY) | | | |
|     <5000 CNY | −1.25 | (−2.70, 0.19) | 0.089 |
|     ≥10000 CNY | −2.76 | (−5.29, −0.22) | 0.033[*] |
| Presence of chronic diseases (ref: no) | −2.24 | (−3.54, −0.93) | <0.001[***] |
| BMI (kg/m$^2$ ) | 0.01 | (−0.19, 0.21) | 0.928 |
| Smoking status (ref: never) | | | |
|     Current | −2.37 | (−5.14, 0.40) | 0.094 |
|     Ceased | −0.15 | (−3.19, 2.88) | 0.922 |
| Alcohol consumption (ref: never) | | | |
|     Current | −0.48 | (−1.98, 1.02) | 0.529 |
|     Ceased | −0.12 | (−2.07, 1.84) | 0.906 |
| Sleep quality (ref: good) | −6.90 | (−8.74, −5.06) | <0.001[***] |
| Physical Activity (ref: low) | | | |
|     Moderate | 5.01 | (3.17, 6.85) | <0.001[***] |
|     High | 5.35 | (3.53, 7.16) | <0.001[***] |

**Notes.**

EQ-VAS, EuroQol-visual analogue scale;  BMI,  Body mass index.

[*]$p < 0.05$
[**]$p < 0.01$
[***]$p < 0.001$

Sex, marital status, and BMI were not significantly associated with any dimension of the EQ-5D-5L descriptive system or the EQ-VAS score.

## DISCUSSION

The results of this study could help to elucidate HRQoL among Chinese seasonal retired migrants in Hainan. The percentages reporting no problems in the first three EQ-5D-5L dimensions (mobility, self-care, and usual activities) were relatively high, whereas the pain/discomfort dimension was the worst among Chinese seasonal retired migrants. Furthermore, this study investigated the potential factors associated with HRQoL.

Consistent with the EQ-5D-5L norms for urban Chinese populations (*Huang et al., 2017*; *Yang et al., 2018*), few participants reported problems in the first three dimensions of the EQ-5D-5L descriptive system (mobility, self-care, and usual activities). However, the percentages of reported problems in the last two dimensions (pain/discomfort and anxiety/depression) among seasonal retired migrants are lower than those among normal urban populations in China. The comparison suggests that seasonal retired migrants have better health status than their nonmigratory contemporaries do in China. Similarly, *Liang,*

*Luo & Hui (2023)* reported that retired Chinese migrants have a high level of subjective well-being, reflecting the high level of happiness and quality of life among retired migrants during their migration. Chinese seasonal retired migrants' high HRQoL could be due to two main explanations: (1) Although the percentages of reporting problems in the first three dimensions of the EQ-5D-5L descriptive system are low among normal Chinese populations, the lifestyle of seasonal migration additionally requires migrants' mobility and self-care ability and usual activities (*Kelly, 2023*), which is consistent with the theory of the "healthy migrant effect". Hence, almost all seasonal retired migrants reported no problems in the first three dimensions. (2) Compared with nonmigratory populations, seasonal retired migrants can escape from cold weather (*Pickering et al., 2019*; *Zhou et al., 2018*). In addition, popular destinations for Chinese seasonal retired migrants usually have better air quality and water quality, more negative oxygen ions, and more greenspace than their original locations do (*Chen & Wang, 2022*). This superb environment is therapeutic and can benefit migrants' health (*Doughty, 2018*). Additionally, the warm and pollution-free environment makes it possible for seasonal retired migrants to exercise outdoors during the winter, which would be impossible if they spent the winter in their locations of origin (*Chen & Wang, 2022*).

Consistent with the findings of a previous study (*Zhang, Zhang & Xu, 2021*), the prevalence of chronic diseases among participants in the current study was lower than that among the nationwide retired population (*Yao et al., 2019*). However, the presence of chronic diseases remained one of the most important factors in the current study because it was significantly correlated with three dimensions of the EQ-5D-5L descriptive system and the EQ-VAS, with a relatively large effect size. These findings are consistent with the findings of previous studies (*Sun et al., 2011*; *Tan et al., 2013*). Given that approximately one-third to one-half of these individuals suffer from chronic diseases (*Zhang, Zhang & Xu, 2021*), seasonal retired migrants with chronic diseases should not be easily neglected when designing health promotion programs.

However, inconsistent with studies on HRQoL (*Poder, Carrier & Kouakou, 2020*; *Szende et al., 2023*), demographic characteristics, including age, sex, and marital status, were not significantly associated with any dimension of the EQ-5D-5L descriptive system in the current study. Such associations could be masked by the potential therapeutic effects of the superb environment mentioned above, which seasonal retired migrants experienced during their migration.

Intriguingly, although the seasonal retired migrants' responses to the EQ-5D-5L descriptive system were not inferior to the EQ-5D-5L norms (*Yang et al., 2018*), their EQ-VAS scores were lower. These results indicate that the participants' self-rated health was lower than that of urban older adults. The disparity in the EQ-VAS scores suggests that seasonal retired migrants perceived a greater distance between their current health and best imaginable health. Specifically, lower self-rated health could result from higher standards for "the best imaginable health status" (*Dowd & Zajacova, 2010*). To partially address the disparity between their current health status and their best imaginable health, Chinese seasonal retired migrants adopt this lifestyle of seasonal migration, with the hope of improving their physical health (*Kou, Xu & Kwan, 2018*).

A cost-effective method to improve HRQoL is lifestyle intervention (*Eriksson et al., 2010*; *Munro et al., 2004*). In the present study, sleep quality was associated with all dimensions of the EQ-5D-5L descriptive system and the EQ-VAS, with relatively large effect sizes. In addition, given that a fair number of seasonal retired migrants suffer from poor sleep quality, providing sleep quality improvement programs to improve their HRQoL is essential. Another important factor found in the current study is physical activity. As mentioned above, seasonal migration could provide migrants with a better environment suitable for outdoor exercise during the winter. Accordingly, the proportion of moderate- and high-level physical activity reached more than 80% in the current study, a value that is considerably higher than that noted among the general Chinese older population (*Zhu, Chi & Sun, 2016*). However, a substantial proportion of seasonal migrants reported low-level physical activity. In addition, as higher-levels of physical activity in this study was negatively associated with the frequency of reporting problems in multiple dimensions of the EQ-5D-5L descriptive system and the EQ-VAS, it is necessary to provide physical activity programs for seasonal retired migrants to improve their HRQoL.

Although obesity and smoking are common risk factors for HRQoL in the general population (*Jia & Lubetkin, 2010*), we could not identify any significant associations between HRQoL and BMI or smoking status (current *vs.* never) in this study. These findings may be attributed to the high homogeneity of BMI and smoking status in the current study sample. Given the relatively small association between alcohol consumption and HRQoL and the inconclusive association between alcohol consumption and HRQoL reported in previous studies (*Imtiaz et al., 2018*; *Ortolá et al., 2016*), providing alcohol abstinence programs to seasonal retired migrants seems unnecessary. In general, on the basis of the current findings, it is essential to provide sleep quality improvement and physical activity programs to seasonal retired migrants to improve their HRQoL, whereas weight loss, smoking cessation or alcohol abstinence programs are less necessary.

These findings provide evidence on Chinese seasonal retired migrants' HRQoL and identify several important associated factors, which enrich the understanding of retired populations with special and increasingly popular lifestyles. In addition, the careful design of the inclusion and exclusion criteria ensures a focused analysis of seasonal retired migrants. However, this study also has several limitations. First, owing to the nature of the cross-sectional design, causal effects between behaviors and HRQoL could not be identified among seasonal retired migrants on the basis of the current study. However, many studies have discussed casual relationships in other populations. Second, our investigation was geographically limited to Wuzhishan city, potentially limiting the generalizability of our findings. Therefore, future larger-scale longitudinal studies are needed for a more comprehensive understanding of this increasing population. Third, the HRQoL of seasonal retired migrants was not compared with that of a control group of non-migrants in our study. As a remedy measure, we discussed the difference in HRQoL between our sample and national EQ-5D-5L norms and discovered a high level of HRQoL among Chinese seasonal retired migrants in Hainan. Fourth, other potential factors associated with HRQoL were not explored in the current study. For example, we did not test the association between

migrants' HRQoL and their adjustment to the local society, which could influence migrants' mental health (*Zhong et al., 2016*).

This study overlapped with the COVID-19 pandemic, which may affect both migration (*Guadagno, 2020*) and HRQoL (*Poudel et al., 2021*). However, the effects of COVID-19 on migration are largely due to mobility restrictions. In late 2022, China lifted its mobility restrictions, resulting in a population movement scale similar to that of 2019, before the COVID-19 pandemic (*Zhang et al., 2023*). Further research is needed to understand the long-term trends in the health of seasonal retired migrants in China.

## CONCLUSIONS

This study investigated the HRQoL of Chinese seasonal retired migrants in Hainan Province and explored the factors associated with HRQoL. The results revealed that seasonal retired migrants have a high level of HRQoL and that pain/discomfort is the major problem for these migrants. Additionally, this study revealed that the presence of chronic diseases, poor sleep quality, and low-level physical activity are the most important factors negatively associated with HRQoL because they are related to multiple dimensions of the EQ-5D-5L descriptive system and EQ-VAS score, with relatively high effect sizes. These findings can provide a more comprehensive understanding of this unique population and are helpful for policy-makers in designing tailored policies to improve the HRQoL of seasonal retired migrants.

On the basis of the current findings, chronic conditions, which require sufficient health care services, should never be neglected. Furthermore, as sleep quality and physical activity are also important associated factors, behavioral intervention programs for sleep quality improvement and physical activity represent practical choices, whereas interventions for other aspects, such as smoking, alcohol consumption, and weight loss, are less necessary. Future longitudinal studies are needed to compare the health trajectories of seasonal retired migrants with those of nonmigratory populations with similar socioeconomic statuses, and information from these studies could further elucidate the impact of seasonal migration on health status.

### Funding

This work was supported by the Wuzhishan Municipal Health Commission. The funders had no role in study design, data collection and analysis, decision to publish, or preparation of the manuscript.

### Grant Disclosures

The following grant information was disclosed by the authors:
Wuzhishan Municipal Health Commission.

### Competing Interests

Jinming Yu is an Academic Editor for PeerJ. Lin Ouyang is employed by WUZHISHAN Snowbird Medical Professionals Workstation.

## Author Contributions

- Sikun Chen conceived and designed the experiments, performed the experiments, analyzed the data, prepared figures and/or tables, authored or reviewed drafts of the article, and approved the final draft.
- Tianchang Li conceived and designed the experiments, performed the experiments, authored or reviewed drafts of the article, and approved the final draft.
- Lingjun Wang conceived and designed the experiments, performed the experiments, authored or reviewed drafts of the article, and approved the final draft.
- Shigong Wang conceived and designed the experiments, performed the experiments, authored or reviewed drafts of the article, and approved the final draft.
- Lin Ouyang conceived and designed the experiments, performed the experiments, authored or reviewed drafts of the article, and approved the final draft.
- Jiwei Wang conceived and designed the experiments, analyzed the data, prepared figures and/or tables, authored or reviewed drafts of the article, and approved the final draft.
- Dayi Hu conceived and designed the experiments, authored or reviewed drafts of the article, and approved the final draft.
- Jinming Yu conceived and designed the experiments, authored or reviewed drafts of the article, and approved the final draft.

## Human Ethics

The following information was supplied relating to ethical approvals (*i.e.*, approving body and any reference numbers):

Ethics approval was obtained from the medical ethics committee of School of Public Health, Fudan University (IRB00002408 & FWA00002399, approval number IRB#2022-02-0944).

## Data Availability

The raw data and corresponding SAS code are available in the Supplementary File.

## Supplemental Information

Supplemental information for this article can be found online at http://dx.doi.org/10.7717/peerj.18574#supplemental-information.

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
