# Peer review of "Health-related quality of life and its associated factors among Chinese seasonal retired migrants in Hainan"

_PeerJ, doi:10.7717/peerj.18574_

## Round 0.1 · original submission · Major Revisions

While this study has its merits, certain issues must be addressed before advancing its progress.

Reviewer 1 ·

Basic reporting

The cross-sectional survey focuses on HRQOL among Chinese seasonal retired migrants in Wuzhishan, Hainan province, China. Given the limited data available, it is important to document these findings.

First, ensure that your title accurately limits the scope to the specific region mentioned.

Second, provide the extended form of EQ-5D-5L in both the main text and abstract upon its initial usage.

Third, consider incorporating a brief review of the healthy migrant effect and the salmon bias hypothesis in the introduction, linking these theories to the health of Chinese seasonal retired migrants (Refer: PMID: 23508367, PMID: 37456617, PMID: 35926662). This would enhance the readers' interest!

Fourth, a brief review of similar HRQOL studies, such as those involving rural-to-urban migrant workers, could further enrich your paper (Refer: PMID: 25403568).

Fifth, your paper's English language usage requires additional editing for clarity. For example, specify the year when there was an increase of a million in line 63. Also, in line 67, identify the 'non-migrants' as either locals or origin people, as the comparative results could differ.

Sixth, health status and HRQOL don't equate; a review on health status in seasonal retired migrants is therefore misplaced in your introduction. Please provide further elucidation on this.

Seventh, as your study period is from October to November 2022, consider the potential influence of the COVID-19 pandemic.

Eighth, assessing migrants' adjustment to the local society is significant. However, it appears this was overlooked in your methodology (Refer: PMID: 27300005).

Ninth, it's essential to mention the limitation of not having a control group of non-migrants, either locals or people of the same origin. Consequently, temper the early conclusions drawn from your study.

Finally, it's noteworthy that the acceptable cut-off score for the Chinese PSQI is 7, not 5.

Experimental design

The cross-sectional survey focuses on HRQOL among Chinese seasonal retired migrants in Wuzhishan, Hainan province, China. Given the limited data available, it is important to document these findings.

First, ensure that your title accurately limits the scope to the specific region mentioned.

Second, provide the extended form of EQ-5D-5L in both the main text and abstract upon its initial usage.

Third, consider incorporating a brief review of the healthy migrant effect and the salmon bias hypothesis in the introduction, linking these theories to the health of Chinese seasonal retired migrants (Refer: PMID: 23508367, PMID: 37456617, PMID: 35926662). This would enhance the readers' interest!

Fourth, a brief review of similar HRQOL studies, such as those involving rural-to-urban migrant workers, could further enrich your paper (Refer: PMID: 25403568).

Fifth, your paper's English language usage requires additional editing for clarity. For example, specify the year when there was an increase of a million in line 63. Also, in line 67, identify the 'non-migrants' as either locals or origin people, as the comparative results could differ.

Sixth, health status and HRQOL don't equate; a review on health status in seasonal retired migrants is therefore misplaced in your introduction. Please provide further elucidation on this.

Seventh, as your study period is from October to November 2022, consider the potential influence of the COVID-19 pandemic.

Eighth, assessing migrants' adjustment to the local society is significant. However, it appears this was overlooked in your methodology (Refer: PMID: 27300005).

Ninth, it's essential to mention the limitation of not having a control group of non-migrants, either locals or people of the same origin. Consequently, temper the early conclusions drawn from your study.

Finally, it's noteworthy that the acceptable cut-off score for the Chinese PSQI is 7, not 5.

Validity of the findings

The cross-sectional survey focuses on HRQOL among Chinese seasonal retired migrants in Wuzhishan, Hainan province, China. Given the limited data available, it is important to document these findings.

First, ensure that your title accurately limits the scope to the specific region mentioned.

Second, provide the extended form of EQ-5D-5L in both the main text and abstract upon its initial usage.

Third, consider incorporating a brief review of the healthy migrant effect and the salmon bias hypothesis in the introduction, linking these theories to the health of Chinese seasonal retired migrants (Refer: PMID: 23508367, PMID: 37456617, PMID: 35926662). This would enhance the readers' interest!

Fourth, a brief review of similar HRQOL studies, such as those involving rural-to-urban migrant workers, could further enrich your paper (Refer: PMID: 25403568).

Fifth, your paper's English language usage requires additional editing for clarity. For example, specify the year when there was an increase of a million in line 63. Also, in line 67, identify the 'non-migrants' as either locals or origin people, as the comparative results could differ.

Sixth, health status and HRQOL don't equate; a review on health status in seasonal retired migrants is therefore misplaced in your introduction. Please provide further elucidation on this.

Seventh, as your study period is from October to November 2022, consider the potential influence of the COVID-19 pandemic.

Eighth, assessing migrants' adjustment to the local society is significant. However, it appears this was overlooked in your methodology (Refer: PMID: 27300005).

Ninth, it's essential to mention the limitation of not having a control group of non-migrants, either locals or people of the same origin. Consequently, temper the early conclusions drawn from your study.

Finally, it's noteworthy that the acceptable cut-off score for the Chinese PSQI is 7, not 5.

Additional comments

None.

·

Basic reporting

Major Comments
Abstract
1. Despite its increasing popularity, the evidence on Chinese seasonal migrants. health status is controversial. What makes it controversial? Kindly expatiate. [Line 28-Line 29]

2. Descriptive analyses of the HRQoL profiles, multiple logistic regression models for the factors associated with each dimension of the EQ-5D, and multivariable linear regression model for the factors associated with the EQ visual analogue scale (EQ-VAS) were performed – What is the difference between multiple logistic regression models and multivariable linear regression model? Please, kindly differentiate them and revise them accordingly [Line 36-Line 38]

Introduction
3. The introduction did not include much vital information on the subject matter. The introduction is the place to review other conclusions on the topic. The introduction section should introduce past findings to those who might not have that expertise. The introduction of this study should follow the ‘Funnel Approach’ to scope and review existing studies globally and the study setting. The ‘funnel approach’ is a method of structuring a literature review that moves from the general to the specific. It is a useful way to organize the literature review when one starts with the broader aspects of the topic and gradually narrowing the focus to the specific aspects of the topic, that one makes the author(s) addressing the subject matter in the study. Therefore, this will illustrate the trends, pattens and prevalence rates of the study outcome of interest of this study. Kindly revise. [Line 56 – Line 97]

4. Where are the gaps identified in this study? What has the previous studies have mentioned as regards the outcome(s) of this study? What is the current study saying on the current situation of your study outcome(s)? Always remember that a gap in a research study is an unanswered question or unresolved problem in a field, which reflects a lack of existing research in that space or it can exist when there is a concept or new idea that hasn't been studied at all, when the existing research is outdated or insufficient, or when there is a disagreement, contextual or methodological issue among the researchers. A gap in a research study indicates an opportunity for further investigation and contribution to the scientific knowledge. Kindly revise. [Line 56 – Line 97]

5. Where are the main objective of this study? Where are the specific objectives of this study? Kindly state them clearly. [Line 56 – Line 97]

Experimental design

Material and Methods

6. Methods should follow in sequence – Study setting and design, sampling method, study population and sampling technique, variable measurements, Ethical considerations, and statistical analysis.

7. Study setting and design - Give a detailed description of the Study setting and design - First, give us a detailed geographical description of the study setting you carried out this study. What is the justification for the choice of the study setting? Also, second, what is the type of design you employed in this study? Kindly address this.

8. Sampling method – In this section, there should be explanation on the study method and, detailed explanations on the methods on the specific tools and procedures use in collecting and analyzing the data should be clarified. Give a justification for the choice of study method employed. Kindly address this.

9. What is HRQoL instrument? Explain in detail what it is and how it has been used in existing studies? No one understands HRQoL instrument and how the tool has been used in other studies? What were the advantages and limitations of HRQoL instrument? Kindly revise. [Line 128 – Line 138]

10. What is Chinese version of EQ-5D-5L? Explain in detail what it is and how it has been used in existing studies? No one understands Chinese version of EQ-5D-5L instrument and how the tool has been used in other studies? What were the advantages and limitations of Chinese version of EQ-5D-5L instrument? Kindly revise. [Line 128 – Line 138]

11. What is EQ-5D descriptive system? Explain in detail what it is and how it has been used in existing studies? No one understands EQ-5D descriptive system and how the tool has been used in other studies? What were the advantages and limitations of EQ-5D descriptive system? Kindly revise. [Line 128 – Line 138]

12. What is EQ visual analogue scale (EQ-VAS)? Explain in detail what it is and how it has been used in existing studies? No one understands EQ visual analogue scale (EQ-VAS) and how the tool has been used in other studies? What were the advantages and limitations of EQ-5D descriptive system? Kindly revise. [Line 128 – Line 138]

13. What do you mean ‘potential factors’ associated with HRQoL? What are the potential factors and why are you using the word ‘potential factors'? [Line 139 – Line 159]. Kindly revise.

14. Study population and Sampling technique – describe your study population? How did you arrived at reaching out to the population target? Give justification of the choice of these procedures you employed? How did you select them? Give us an adequate and precise procedures you used in getting your population. Discuss in detail both of them with a separate section paragraph for each of them. Insert justification for both differently. Kindly address this.

15. What is online Epitools (Ausvet 2024)? How did you used it to calculate the sample size based on participants EQ-VAS scores?

16. Determination of sample size – how did you arrive at 385 sample size for your study? Show it and tell us the sample size determination formula you adopted and give justification for it. How did you arrive at 992 when you have 385 for sample size? Kindly address this. [Line 160 – Line 167] and [Line 192 – Line 193].

17. Discuss the required coefficients for sample size calculations regarding population standard deviation, confidence level, and desired precision. Kindly address this. [Line 162 – Line 163].

18. Provide a detailed information on how you design the research instruments (questionnaires) and show the measurement guidelines of the questionnaire as the instrument for data collection used in this study. Who propounded these instruments? References them and cite studies that have used them to measure the same outcome of interest as it is in your study (either by adopting or adapting).

19. Describe briefly and precisely step by steps, how you used the above adopted or adapted measurement instruments/guidelines to design your study instruments. Show how your variables were measured in the questionnaire.

20. Variable measurements (Outcome variable or Dependent variable) – A dependent variable is the variable that changes as a result of the independent variable (independent or explanatory variables or factors) manipulation. It's the outcome you're interested in measuring, and it “depends” on your independent variable. In statistically analysis, dependent variables are also called ‘response variables’ (they respond to a change in another variable). Therefore, specify your outcome variable very well and show us how it is going to be measured. Is it a binary outcome variable or what? This section should discuss the measurements of the outcome variable or dependent variable(s). Kindly address this.

21. Explanatory Variable or Independent Variable: You did not indicate your independent variables you used in this study. I know that you cannot have the dependent variable without the independent variable when running the statistical analysis. Insert the independent variables and how they are going to be measured (References may be cited). Define all your Explanatory Variables or Independent Variables and how they were measured in your study. This is very important, as it will affect the process and approach of statistical analysis. Kindly address this.

22. How did you measure demographic and behavioral characteristics, BMI, and presence of chronic diseases in this your study? Explain each of the measurements in a separate paragraph under the section Statistical analysis and named this section ‘Variable Measurements’. This should not be included in the statistical analysis section. Kindly address this. [Line 178 – Line 179].

23. Statistical analysis – State and describe the type of statistical analysis suitable for each objective and how they will be analysed in this study. For instance, objective one on respondents’ demographic characteristics was analyzed using univariate analysis (Percentage and frequency including graphs and tables). Objective two measure the associations between the outcome variable and explanatory variables by using the bivariate analysis (Chi-Square or/and Pearson correlation coefficient). Objective three measures the relationship influence between the variables and the predictors of the outcome variable on the explanatory factors, when other factors are kept constant (for instance, when using Logistics regression, if and only if the outcome variable is binary, Yes = 1 and No = 0), otherwise, multiple régression analysis, etc. This section must be revised as it is very important. Also, include the models and equations of the bivariate and inferential statistical analysis in your discussion under the section of the statistical analyses. Kindly address this question.

24. How did your account for missing variables or non-responses from the respondents? Kindly look into this.

25. Where is the “Declaration of Helsinki Ethical principles” of this study? Where is the Ethical Approval? What ethical principles did you employed? Did you follow the set of ethical principles. Where is the informed consent form in Research? How did you address these factors associated with research such as in ensuring in the protection of participants’ rights, ensuring that participants understand the nature and purpose of the research, the risks and benefits of participating, and their rights as participants whether to participate or decline at any time they feel not to. Kindly address this. Was informed consent forms given to the respondents during the data collection?

Validity of the findings

Results
26. Kindly report the results according to the objectives by using the specific objectives as theme heading. Show the type of analysis employed and elaborate on the details of the study findings. Therefore, kindly revise. For instance, see below

- Table 1 – Demographic Characteristics of Respondents
- Table 2 – Correlation coefficients
- Table 3 – Regression coefficients
- Other relevant tables (if any). Kindly address this.

All the interpretations for the Tables should be included in the results and Table legend should be carefully and appropriately inserted in the main research. Interprete and discuss the relevant and significant variables and make it precise so that it doesn’t lose its meaning. Your readers will see the tables with the results. Work on your Tables and make them represent the analyses that were carried in this study. Kindy address these concerns mentioned. [Line 190 – Line 230]

Discussion
27. The discussion section is one of the final parts of a research paper, in which an author describes, analyzes, and interprets their findings. They explain the significance of those results and tie everything back to the research question(s). The discussion section is where you delve into the meaning, importance, and relevance of your results. It should focus on explaining and evaluating what you found, showing how it relates to your literature review and paper or dissertation topic, and making an argument in support of your overall conclusion. Therefore, let your discussion aspects focused on the interpretation of your findings and relate to literature whether it corroborates or not, with existing studies, alongside with in-citation of recent references. Let each paragraph relates the objective findings. Discuss in assertion or not with study’s findings from other studies. Kindly advise.

Conclusion
28. The conclusion should stem from summing up your research paper (by the following steps - restate your research topic, restate the thesis, summarize the main points, state the significance or results, and conclude your thoughts). Always keep in mind that a conclusion is not merely a summary of the main topic(s) covered or a re-statement of your research problem, but a synthesis of key points and, if applicable, where you should recommend new areas for future research.

Recommendations
29. Where is the Recommendation for this study? You can add it to the Conclusion section with the subheading – Conclusion/Recommendations.

Additional comments

References
30. Does this References of this study follow this Journal format? Kindly check and revisit them all.

31. Remove old references; there are recently published articles that have addressed the outcome of interest of this study.

Editing of the English content of the Manuscript
32. Kindly stick to British or American English…you should use one of them.

33. Ask for a Professional English Editors to address all grammatical errors and lexis structure of your manuscript.

Reviewer 3 ·

Basic reporting

Please find my comments in the attachment.

Experimental design

Please find my comments in the attachment.

Validity of the findings

Please find my comments in the attachment.

Additional comments

Please find my comments in the attachment.

Annotated reviews are not available for download in order to protect the identity of reviewers who chose to remain anonymous.

---

## Round 0.2 · accepted · Accept

The authors have addressed the reviewers' concerns.

Reviewer 1 ·

Basic reporting

None

Experimental design

None

Validity of the findings

None

Additional comments

None

Reviewer 3 ·

Basic reporting

The authors have well addessed my comments.

Experimental design

The authors have well addessed my comments.

Validity of the findings

The authors have well addessed my comments.

Additional comments

The authors have well addessed my comments.